# Neural fields for tissue attenuation curve reconstruction in sparsely sampled time-resolved CT

**Lucas de Vries**[1,2,3]       LUCAS.DEVRIES@AMSTERDAMUMC.NL
**Rudolf L.M. van Herten**[2,3]       R.L.M.VANHERTEN@AMSTERDAMUMC.NL
**P. Matthijs van der Sluijs**[4]       P.VANDERSLUIJS@ERASMUSMC.NL
**Ivana Išgum**[1,2,3]       I.ISGUM@AMSTERDAMUMC.NL
**Bart J. Emmer**[1]       B.J.EMMER@AMSTERDAMUMC.NL
**Charles B.L.M. Majoie**[1]       C.B.MAJOIE@AMSTERDAMUMC.NL
**Henk A. Marquering**[*1,2]       H.A.MARQUERING@AMSTERDAMUMC.NL
**Efstratios Gavves**[*3]       E.GAVVES@UVA.NL

[1] *Amsterdam UMC, location University of Amsterdam, Radiology and Nuclear Medicine, Meibergdreef 9, Amsterdam, 1105 AZ, The Netherlands*
[2] *Amsterdam UMC location University of Amsterdam, Biomedical Engineering and Physics, Meibergdreef 9, Amsterdam, 1105 AZ, The Netherlands*
[3] *Informatics Institute, University of Amsterdam, Amsterdam, The Netherlands*
[4] *Department of Radiology and Nuclear Medicine, Erasmus MC, University Medical Center Rotterdam, Rotterdam, The Netherlands*

**Editors:** Accepted for publication at MIDL 2025

## Abstract

Time-resolved CT imaging can aid acute ischemic stroke diagnosis by visualizing contrast agent transport through the brain (micro)vasculature. CT perfusion imaging, while widely used for stroke diagnosis, requires approximately 30 sequential scans, leading to extensive radiation exposure and motion sensitivity. As an alternative to CT perfusion imaging, some hospitals opt for multiphase CT angiography for time-resolved analysis with reduced radiation dose. However, multiphase CT angiography lacks standardized perfusion analysis capabilities, making it more challenging to interpret than CT perfusion imaging. We present Sparse Temporal Attenuation Reconstruction (STAR), a novel approach using conditional neural fields that reconstructs tissue attenuation curves from sparse observations, allowing for reduced radiation exposure and motion sensitivity with CT perfusion, while enabling perfusion analysis from multiphase CT angiography. Our method generates full tissue attenuation curves using only 4 out of 30 observations. The results show that perfusion maps from reconstructed data match the reference perfusion maps, potentially reducing radiation and allowing recovery of motion-corrupted images. Moreover, STAR enables perfusion analysis in centers using multiphase CT angiography. Consequently, STAR has the potential to improve the stroke imaging work-up while making perfusion analysis more widely accessible.

**Keywords:** conditional neural fields, CT perfusion, multiphase CT angiography, acute ischemic stroke

---

* Contributed equally

## 1. Introduction

Assessment of hemodynamic consequences of stroke is commonly performed by time-resolved CT imaging such as CT perfusion or multiphase CT angiography, where the transport of the contrast agent through the intracranial (micro) vasculature is assessed. CT perfusion is widely regarded as the standard method for the hemodynamic analysis in most hospitals. This technique involves acquiring approximately 30 CT scans at 1–3 second intervals after a contrast agent is administered. The resulting images are subsequently analyzed by software to generate *perfusion maps*, which summarize tissue hemodynamics such as blood flow and transit times. However, CT perfusion faces several limitations in clinical practice. Firstly, each acquisition exposes the patient to radiation that builds up to a significant radiation dose with 30 acquisitions. Secondly, patient movement can cause severe motion artifacts in acquisitions (Fahmi et al., 2013). If a significant number of images is affected by motion artifacts, the CT perfusion analysis fails. Previously, multiphase CT angiography has been introduced to deal with extensive radiation exposure (Menon et al., 2015). Multiphase CT angiography extends CT angiography with two additional acquisitions, offering temporal information with minimal protocol changes and little additional costs and scan time (Menon et al., 2015; Dundamadappa et al., 2021). However, multiphase CT angiography does not allow for a straightforward generation of perfusion maps, which have superior sensitivity for detecting perfusion defects, particularly for distal occlusions (Benali et al., 2023).

We propose an approach that tackles CT perfusion's main drawbacks: radiation dose and motion sensitivity. Furthermore, we show that our approach allows for performing perfusion analysis utilizing multiphase CT angiography data, all while maintaining compatibility with commercial off-the-shelf perfusion analysis software. The core of our method lies in reconstructing the complete temporal evolution of contrast enhancement, also known as tissue attenuation curves, using only a limited number of measurements. Previous research focused on the interpolation of uniformly undersampled CT perfusion data to reduce radiation (Bae et al., 2024). Others used multiphase CT angiography acquisitions to estimate perfusion parameter maps through methods such as calculating the per-voxel slope through three attenuation measurements (McDougall et al., 2020) or diffusion models (Cai et al., 2024). However, these methods are unable to extrapolate beyond acquisitions, restricted to interpolation between temporally close samples (Bae et al., 2024), and focused on generating software-specific perfusion maps (Son et al., 2024; Cai et al., 2024) rather than reconstructing tissue attenuation curves, which can be used with any analysis software.

We propose STAR: Sparse Temporal Attenuation Reconstruction, which learns continuous voxel-wise temporal representations of attenuation curves using conditional neural fields to enable attenuation curve reconstruction from sparse measurements. Consequently, STAR can estimate the tissue attenuation curve from only 4 out of 30 observations (13%). These four acquisitions correspond to time points that align with non-contrast CT and multiphase CT angiography acquisition protocols. The generated attenuation data can subsequently be analyzed with off-the-shelf CT perfusion analysis software. We show that the obtained perfusion maps are on par with the perfusion maps obtained from the fully sampled CT perfusion source data. STAR allows for perfusion analysis of attenuation data with 87% fewer CT acquisitions, which could lead to a significant radiation reduction. Additionally, this method could be used to reconstruct parts of the sequence when a significant number

of acquisitions are corrupted by motion artifacts. Furthermore, we demonstrate that STAR can reconstruct sequential attenuation data using non-contrast CT and multiphase CT angiography measurements, enabling perfusion analysis based on multiphase CT angiography.

## 2. Method

In the following, we introduce STAR. Figure 1 presents a visual overview of the method.

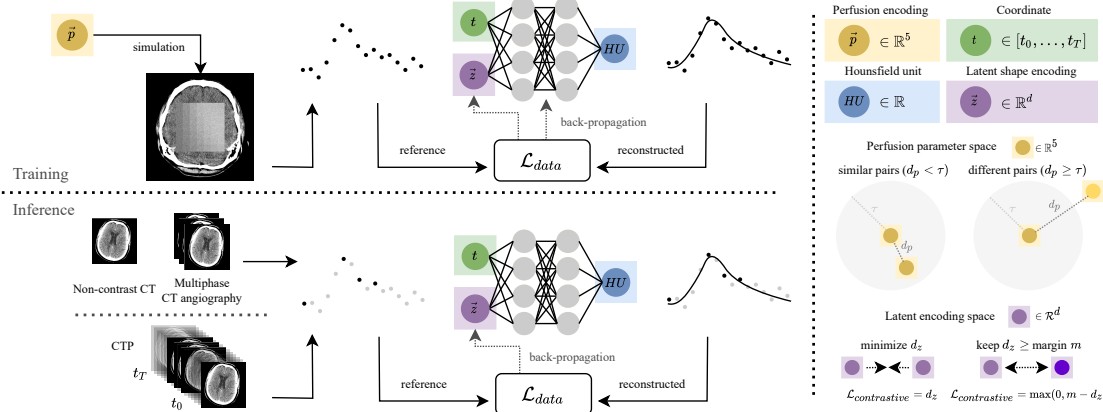

Figure 1: Overview of STAR. During training, a conditional neural field is trained on fully sampled simulated CT perfusion data with simulation parameters $\boldsymbol{p} \in \mathbb{R}^5$ (CBF, CBV, MTT, TMAX, DELAY). At inference, only the latent variable is updated based on the observed data on the subsampled domain.

### 2.1. Conditional neural fields

In physics, a field $f$ is defined as a scalar or vector quantity over a particular domain. For example, we can view the dynamic contrast attenuation as a field defined on the temporal domain. A *neural* field is a field represented by a neural network $f_\theta$ with parameters $\theta$ (Xie et al., 2022). In our method, we use the simplest one-dimensional and scalar form of a neural field $f_\theta : \mathbb{R} \to \mathbb{R}$, because we want to learn the neural field of the scalar attenuation (given in Hounsfield Units) only on the temporal domain $t \in [t_0, ..., t_T]$. The goal is to infer the complete tissue attenuation curve using only limited measurements. Since attenuation curves often share similar patterns, training a separate neural field for each curve fails to leverage these shared characteristics. Moreover, training individual neural fields is computationally inefficient, as each curve requires its own complete training process. Therefore, we employ *conditional* neural fields.

In a conditional neural field, we enhance the neural field by adding a $d-$dimensional latent variable $\boldsymbol{z} \in \mathbb{R}^d$ as input, allowing us to represent many different curves with a single field $f_\theta : \mathbb{R}^{1+d} \to \mathbb{R}$. This latent variable $\boldsymbol{z}$ is essentially an embedding of the curve's shape or a *shape conditional*. In practice, a latent $\boldsymbol{z}$ exists for each instance in the data set.

During training, we optimize both the neural field parameters $\theta$ and the latent variables $\boldsymbol{z}$ to minimize the difference between our predictions and the training data. At inference, we keep the neural field parameters $\theta$ fixed and iteratively adjust the latent variable $\boldsymbol{z}$ to minimize the difference between the neural field predictions and the available data points for a particular number of iterations. The attenuation data are not required to be densely sampled on the temporal domain: we may employ a sub-sampled set. We thus optimize the latents $\boldsymbol{z}$ with data points from the sampled domain. Once we have the optimized latent variable, we can reconstruct the complete tissue attenuation curve by sampling $f_\theta(t, \boldsymbol{z})$ at all $t \in [t_0, ..., t_T]$, allowing us to fill the gaps in the temporal data.

**Training** All latent variables $\boldsymbol{z} \in \mathbb{R}^d$ are randomly initialized from $\mathcal{N}(0, \frac{1}{\sqrt{d}})$, where $d = 32$ (Park et al., 2019), which promotes effective optimization and avoids regions with vanishing gradients. The network $f_\theta(t, \boldsymbol{z})$ consists of a single hidden layer of dimension 128 with tanh activations and outputs the predicted value through a final sigmoid activation function. We use a compound loss function consisting of three losses. The data reconstruction loss $\mathcal{L}_{\text{data}}$ is a $\ell_2$-loss based on the difference between our predictions and the training data with all 30 time points. We use a regularization loss $\mathcal{L}_{\text{reg}} = \sum_i \|z_i\|_2$ to prevent the latents from growing arbitrarily large. Finally, we use a contrastive loss $\mathcal{L}_{\text{contrastive}}$ to ensure that latents corresponding to similar attenuation curves are close together in the latent space. For any two curve samples $i$ and $j$, we measure the similarity of the perfusion parameters $\boldsymbol{p} \in \mathbb{R}^5$, corresponding to the cerebral blood flow (CBF), cerebral blood volume (CBV), mean transit time (MTT), time-to-maximum (TMAX), and the DELAY, that were used to simulate these curves (which we denote as $d_{\boldsymbol{p}}$) and look at how far apart they are in latent space (denoted as $d_{\boldsymbol{z}}$). For similar pairs (when $d_{\boldsymbol{p}} < \tau$), we want their latents to be close, so we directly penalize their distance $d_{\boldsymbol{z}}$. For different pairs (when $d_{\boldsymbol{p}} \geq \tau$), we want their latents to be at least margin $m$ apart. If they are too close, we penalize with $\max(0, m - d_{\boldsymbol{z}})$.

$$\mathcal{L}_{\text{contrastive}} = \frac{1}{|P|} \sum_{i,j} \begin{cases} d_{\boldsymbol{z}}(i,j) & \text{if } d_{\boldsymbol{p}}(i,j) < \tau \text{ (similar pairs)} \\ \max(0, m - d_{\boldsymbol{z}}(i,j)) & \text{if } d_{\boldsymbol{p}}(i,j) \geq \tau \text{ (different pairs)} \end{cases} \tag{1}$$

To make all comparisons fair, $\boldsymbol{p}$ and $\boldsymbol{z}$ are normalized. We average over all possible pairs (the total number of pairs is denoted as $|P|$). The threshold $\tau = 0.1$ decides what we consider similar, while $m = 1.0$ defines how far apart different latents should be. Figure 1 visualizes the contrastive loss. The total training loss, with weights empirically set, is:

$$\mathcal{L} = 100 \cdot \mathcal{L}_{\text{data}} + 0.1 \cdot \mathcal{L}_{\text{reg}} + 10 \cdot \mathcal{L}_{\text{contrastive}} \tag{2}$$

We optimize both the model parameters and the latents using Adam and employ a cosine learning rate schedule that decays the initial learning rate $10^{-2}$ to $10^{-6}$. We train for 16,000 iterations with batches of 4,000 densely sampled attenuation curves. Training takes 22 minutes on consumer hardware with an NVIDIA RTX 2080S GPU and requires $\sim 1.6\,\text{GB}$.

**Inference** All latent variables $\boldsymbol{z}$ are randomly initialized from $\mathcal{N}(0, \frac{1}{\sqrt{d}})$. We optimize the latents $\boldsymbol{z}$ with $\mathcal{L}_{\text{data}}$ and $\mathcal{L}_{\text{reg}}$ with the same weights as during training, using only the observed data while keeping the neural field parameters $\theta$ fixed. We use $\ell_1$-loss during inference (rather than the $\ell_2$-loss used during training) to better preserve the tail characteristics of the attenuation curves. We optimize for 1000 iterations using Adam (<1 minute

and requiring $\sim 0.7$ GB GPU memory) with learning rate $10^{-2}$ with a cosine learning rate schedule decaying the learning rate to $10^{-6}$. To obtain the attenuation values at all time points we infer the network $f_\theta(t, \boldsymbol{z})$ at all $t \in [t_0, ..., t_T]$.

## 2.2. Datasets

We use simulated phantom data to train, and validate with both phantom and patient data. We refer to Appendix A for more details regarding the data and preprocessing.

**Phantom CT perfusion data**  In perfusion analysis, cerebral blood flow (CBF), cerebral blood volume (CBV), mean transit time (MTT), time-to-maximum (TMAX), and the DELAY are the main parameters of interest. The phantom developed by Kudo et al. (2012) included CBV values of 1–5 ml/100g. Here, we expand the phantom to cover a wider range of 1–20 ml/100g such that our method learns to operate on attenuation curves corresponding to non-tissue voxels like vessels. In total, we trained with 735 perfusion parameter combinations.

**Patient CT perfusion data**  We evaluate with CT perfusion data sets from 17 patients from the Ischemic Stroke Lesion Segmentation Challenge (ISLES) 2024 (de la Rosa et al., 2024). We used one patient for validation and the other 16 patients for testing.

**Patient multiphase CT angiography data**  We curated a dataset from Erasmus MC of five patients with matched non-contrast CT, multiphase CT angiography, and CT perfusion.

## 2.3. Baseline: gamma variate model

We fit a voxel-by-voxel gamma variate model to the attenuation curves (Thompson et al., 1964) using constrained curve-fitting. Initialization and bounds are based on the expected contrast dynamics (details provided in Appendix B). The model parameters represent physical properties: peak enhancement, arrival time, rise time, and washout rate. Given the inherent noise in the data, we use this parametric model over more flexible approaches.

## 3. Experiments

**Phantom study**  We compare the perfusion parameters obtained from the complete CT perfusion scan with different subsampling scenarios. The first two scenarios correspond to uniform temporal subsampling at 15/30 ($t \in [t_0, t_2, ...]$) and 7/30 ($t \in [t_0, t_4, ...]$). The third scenario corresponds to time points aligning with multiphase CT angiography acquisition times. These include the first frame $t = t_0$ and frames corresponding to the peak arterial phase (the time point where the arterial input function peaks) and two delayed phases (peak venous and late venous): $t \in [t_{\mathrm{pa}}, t_{\mathrm{pv}}, t_{\mathrm{lv}}]$. The selected time points are at: $t_0 = 0$ seconds (the first frame of the sequence), $t_{\mathrm{pa}} = 16$ seconds, $t_{\mathrm{pv}} = 24$ seconds, and $t_{\mathrm{lv}} = 32$ seconds, using similar intervals between phases as with multiphase CT angiography acquisition protocols (Menon et al., 2015). Lastly, as a fourth scenario, we use $t \in [t_0, t_{\mathrm{pa}}]$. We use commercially available CT perfusion analysis software SYNGO.VIA (version VB60S; Siemens Healthcare, Erlangen, Germany) to obtain the perfusion parameters, and qualitatively and quantitatively compare the perfusion parameters obtained from STAR-reconstructions with the fully sampled CT perfusion data. Moreover, we quantitatively analyze the gamma variate model as a baseline. We calculate the mean absolute error as an evaluation metric.

Table 1: Mean absolute error in perfusion parameter estimation from phantom data.

| Scenario | Total time pts. | CBF [ml/100g/s] | CBV [ml/100g] | MTT [s] | TMAX [s] | DELAY [s] |
|---|---|---|---|---|---|---|
| $t \in [t_0, t_2, ...]$ | 15/30 | 1.6 | 0.2 | 2.1 | 0.9 | 0.6 |
| $t \in [t_0, t_4, ...]$ | 7/30 | 6.4 | 0.3 | 3.7 | 1.6 | 0.8 |
| $t \in [t_0, t_{pa}, t_{pv}, t_{lv}]$ – proposed | 4/30 | 4.5 | 0.3 | 3.8 | 1.6 | 0.8 |
| $t \in [t_0, t_{pa}, t_{pv}, t_{lv}]$ – curve fit | 4/30 | 10.3 | 0.4 | 6.6 | 4.4 | 3.8 |
| $t \in [t_0, t_{pa}]$ | 2/30 | 12.7 | 2.6 | 3.2 | 1.8 | 0.8 |

**Patient study** We extract sparse measurements $t \in [t_0, t_{pa}, t_{pv}, t_{lv}]$ from fully sampled CT perfusion data. Our method is flexible regarding temporal spacing and can handle any distribution of measurements, making it compatible with various CT scanner acquisition protocols. Since, at inference time, the arterial peak is not available, we use the same time points that we extracted from the phantom data. With two commercially available CT perfusion analysis software solutions: SYNGO.VIA and STROKEVIEWER (version 3.2.11; Nicolab, Amsterdam, The Netherlands), we qualitatively compare perfusion maps generated from fully sampled and STAR-derived CT perfusion data from these sparse measurements and assess the volumetric agreement between penumbra and ischemic core volumes.

**Multiphase CT angiography study** As a proof-of-concept, we qualitatively assess agreement between SYNGO.VIA and STROKEVIEWER perfusion maps from STAR-derived data reconstructed from multiphase CT angiography with those from CT perfusion.

**Additional studies** With UMAP (McInnes et al., 2018), a dimensionality reduction technique, we visualize the latent space to assess if physiologically similar curves are encoded proximally in the latent space and if the contrastive loss stimulates this space to be well-structured. Furthermore, as ablation studies, we investigate loss function combinations and $t_{pa} \leftrightarrow t_{pv} \leftrightarrow t_{lv}$ intervals on the phantom data perfusion parameter accuracy.

## 4. Results

**Phantom study** All perfusion parameters derived from STAR reconstructions show strong agreement with those from complete CT perfusion data. Table 1 lists the mean absolute error. We refer to Table 3 in Appendix C for the mean error en standard deviation. As we subsample more extensively, we see that the errors in the temporal perfusion parameters increase. For the scenario with four time points, the gamma variate model performs poorly, with particularly large errors in temporal parameters. Figure 4 in Appendix C confirms the error increase with more sub-sampling by comparing the CBF and TMAX maps.

**Patient study** Figure 2 presents SYNGO.VIA and STROKEVIEWER perfusion maps. We observe strong visual correspondence between the perfusion maps from the STAR-derived and fully sampled CT perfusion data. The main deviations are in the temporal perfusion parameter TMAX. Table 2 lists the volumetric agreement between infarct core and penumbra estimates. When comparing STAR-derived versus fully sampled CT perfusion data, the

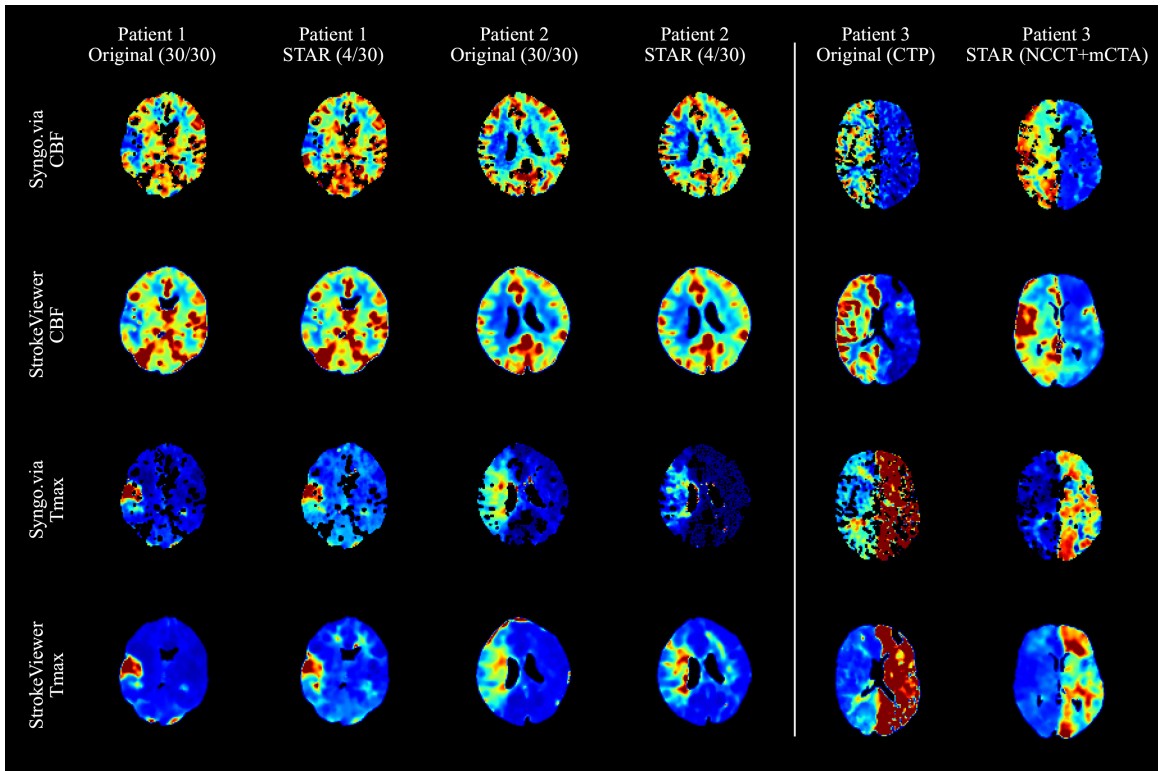

Figure 2: Perfusion maps from both Syngo.via and StrokeViewer comparing: (left) fully sampled CT perfusion versus four timepoints and (right) CT perfusion versus non-contrast CT with multiphase CT angiography.

Table 2: Infarct core and penumbra volume (ml) estimates based on reconstructed and fully sampled data, from Syngo.via and StrokeViewer. Listing median (IQR) (absolute) volumetric difference (VD, AVD) over the test patients. Symbols indicate if smaller ($\downarrow$) or closer to zero (0) values denote better performance.

| Data | Volume | Syngo.via | | StrokeViewer | |
|---|---|---|---|---|---|
| | | VD (0) | AVD ($\downarrow$) | VD (0) | AVD ($\downarrow$) |
| CTP | Core | −5.5 (−18.1–0.1) | 10.8 (3.1–22.7) | 0.0 (−3.5–4.5) | 5.0 (0.0–15.5) |
| | Penumbra | 1.3 (−22.4–28.6) | 28.4 (12.5–47.6) | −1.5 (−11.3–4.8) | 10.5 (4–18.0) |

median (IQR) volumetric differences are –5.5 (–18.1–0.1) ml (Syngo.via) and 0.0 (–3.5–4.5) ml (StrokeViewer) for the infarct core. For reference, Appendix D lists the inter-software differences (StrokeViewer–Syngo.via) for infarct core and penumbra volumes. In the

case of fully sampled CT perfusion data, the difference is –10.5 (–10.2–(–6.5)) ml for the infarct core.

**Multiphase CT angiography study** Figure 2 demonstrates the correspondence between perfusion maps from STAR-derived attenuation data from clinical multiphase CT angiography data and CT perfusion data. We refer to Appendix E for more examples.

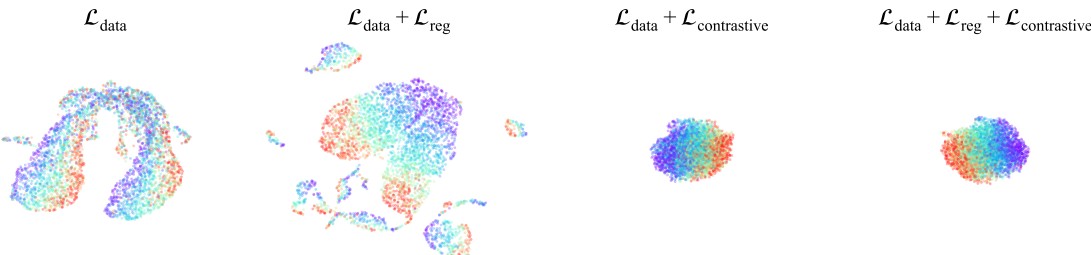

Figure 3: Visualization of the latent space under different loss function combinations. The color of each dot represents its CBV value.

**Additional studies** Figure 3 visualizes the latent space under different loss functions. The figure shows that contrastive loss groups similar perfusion patterns closer together. We refer to Appendix F for more details. The ablation studies in Appendix G show the impact of different loss function combinations and different $t_{\text{pa}} \leftrightarrow t_{\text{pv}} \leftrightarrow t_{\text{lv}}$ intervals on perfusion parameter accuracy on the phantom data, demonstrating that contrastive learning improves perfusion parameter accuracy and that an interval of 6 or 8 seconds is appropriate.

## 5. Discussion and Conclusion

STAR enables a CT perfusion subsampling approach that aligns with the *As Low As Reasonably Achievable (ALARA)* principle. STAR allows a significant reduction in radiation exposure while maintaining diagnostic quality. The high undersampling rate suggests that the method can also reconstruct CT perfusion data with severe motion corruption, allowing for perfusion analysis even in such cases. Moreover, STAR shows promise in reconstructing full-sequence attenuation data from multiphase CT angiography. Since STAR completes CT attenuation curves rather than directly generating perfusion maps, clinicians can use off-the-shelf CT perfusion analysis software without modifying existing workflows.

We note three observations that need discussion. First, we observed deviations in the temporal perfusion maps, as sparse sampling naturally misses exact bolus arrival time and wash-out (Table 1). The deviations increase with more aggressive subsampling protocols. Despite these temporal deviations, our method maintains sufficient clinical value. Second, STAR's training on tissue contrast attenuation curves creates a constraint: the model struggles to reconstruct higher attenuation values typical of arterial and venous structures, despite our expanded phantom's cerebral blood volume range. Consequently, this limitation affects the automated selection of arterial input and venous outflow locations by CT perfusion software, impacting perfusion estimates. Using population-based arterial input and

venous outflow attenuation curves could offer a solution. Finally, replacing CT perfusion attenuation data with the combination of non-contrast CT and multiphase CT angiography data results in increased perfusion parameter deviations. While we can detect infarcts qualitatively, the perfusion maps lack precision needed for accurate core-penumbra volume calculations. Differences in acquisition protocols, tube current, and kilovolt-peak settings impact attenuation values. These findings highlight the need to explore alignment between non-contrast CT and multiphase-CT angiography to match CT perfusion source data.

While Tmax shows expected deviations with 87% fewer acquisitions, the core and penumbra volume measurements remain comparable to fully-sampled data, with differences within inter-software variability ranges. For centers using multiphase CT angiography, our method enables perfusion analysis that would otherwise be unavailable, supporting both radiation safety and diagnostic needs.

star only considers the temporal domain for reconstruction. Future work could incorporate spatial context through neural fields that operate in spatial and temporal domains (Dupont et al., 2022; Bauer et al., 2023), leveraging brain tissue's spatial coherence where neighboring voxels share similar attenuation patterns. However, for complex spatio-temporal signals, neural fields with global conditioning scale poorly (Dupont et al., 2022; Bauer et al., 2023; Papa et al., 2024; Xie et al., 2022). Recent work on equivariant neural fields could provide a solution with geometry-informed latent spaces (Wessels et al., 2024).

We deliberately kept the network architecture small, as our experiments with deeper networks showed they captured high-frequency artifacts rather than the fundamental shape of attenuation curves. The contrastive loss provides further regularization of the latent space, ensuring reconstructed curves maintain physiologically plausible shapes even from very sparse measurements – a clear advantage over the gamma-variate model, which lacks flexibility to capture the full range of attenuation patterns. The threshold $\tau = 0.1$ was chosen to represent approximately 10% difference in normalized parameters, which proved effective in practice, while $m = 1.0$ was an empirical choice without a specific physiological basis. In future work, these parameters could be systematically tuned for potentially even better performance. While the five perfusion parameters suggest a minimum dimensionality for the latent space, our larger latent space provides flexibility to capture complex, non-linear relationships between parameters. Future work could explore other approaches like physics-informed losses to constrain solutions toward physiologically valid curves, particularly when working with extremely limited temporal samples (De Vries et al., 2023).

Our validation cohort is limited in size, which represents a current limitation to be addressed in future work with larger clinical datasets.

While our paper focuses on temporal undersampling, we acknowledge other dose reduction strategies exist, including hardware-based approaches (Lira et al., 2015), iterative reconstruction algorithms (Rapalino et al., 2012), and denoising techniques (Chen et al., 2017), which could be combined with our method for greater reduction.

In conclusion, we demonstrated that reconstructing full perfusion attenuation curves from as few as four measurements is possible, potentially allowing both substantial radiation dose reduction and correction of motion-corrupted acquisitions. Our method enables perfusion measurements from standard multiphase CT angiography acquisition protocols while maintaining compatibility with existing clinical software and workflows.

## Acknowledgments

This work was part of the Artificial Intelligence for Early Imaging-Based Patient Selection in Acute Ischemic Stroke (AIRBORNE) project. This project was supported by Top Sector Life Sciences & Health and Nicolab B.V. We gratefully acknowledge Erasmus MC for providing access to the imaging data used in this study.

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

## Appendix A. Dataset description

**Phantom CT perfusion data**    The phantom includes attenuation curves corresponding to cerebral blood volume: $\text{CBV} \in \{1, 2, 3, 4, 5, 6, 7, 8, 9, 10, 12, 14, 16, 18, 20\}$ ml/100g. The mean transit time and delay perfusion values used for simulation are: $\text{MTT} \in \{3.4, 4.0, 4.8, 6.0, 8.0, 12.0, 24.0\}$ s and $\text{DELAY} \in \{0.0, 0.5, 1.0, 1.5, 2.0, 2.5, 3.0\}$ s. The corresponding cerebral blood flow (CBF) and time-to-maximum (TMAX) are: $\text{CBF} = \text{CBV}/\text{MTT}$ ml/100g/s and $\text{TMAX} = \text{DELAY} + \frac{1}{2}\text{MTT}$ s. The data also includes a simulated arterial input function and a simulated venous output function. All attenuation data is generated with a temporal resolution of two seconds and with a total acquisition time of 58 seconds (30 time points). The attenuation curves are generated through the convolution of an arterial input function with a box-shaped impulse response function. The phantom is organized in a series of axial slices, where each slice represents a distinct CBV level. Within each slice, the data is arranged in a $7 \times 7$ grid of tiles. In this grid, the MTT values increase from right to left, which consequently means that cerebral blood flow ($\text{CBF} = \text{CBV}/\text{MTT}$) increases from left to right. The delay values increase from top to bottom. Each tile contains $32 \times 32$ tissue attenuation curves with different noise realizations. We use a 50/25/25 split for training, validation, and testing. After splitting the data we apply a Gaussian filter with a standard deviation of one voxel to each tile in each axial slice. We scale all attenuation values between 0 and 1. We subtract the attenuation value of the first time point as a baseline and restore the baseline after inferring the attenuation curve.

**Patient CT perfusion data**    We resampled the CT perfusion data sets from 17 patients from the Ischemic Stroke Lesion Segmentation Challenge (ISLES) 2024 (de la Rosa et al., 2024) to a temporal resolution of 2 seconds and motion-corrected all CT perfusion images by registering all frames to the first frame of the sequence. Thereafter, we applied the bilateral smoothing filter ($\sigma_{\text{domain}} = 3$ mm, $\sigma_{\text{range}} = 10$ HU) to approximate the signal-to-noise of the phantom data.

**Patient non-contrast and multiphase CT angiography data**    The proof-of-concept dataset from Erasmus MC consisted of five patients with matched non-contrast CT, multiphase CT angiography, and CT perfusion source data. The patients participated in the MR CLEAN-NO IV trial (Treurniet et al., 2021). The MR CLEAN-NO IV trial (MEC-2017-368) obtained ethics approval from Erasmus MC University Medical Center in Rotterdam and required written informed consent from all participants. The CT perfusion data were temporally resampled to 2-second intervals to maintain consistency with our previous experiments. We applied identical preprocessing steps as described above, including motion correction through registration to the first frame and bilateral filtering. All non-contrast CT and multiphase CT angiography volumes were spatially aligned to the first frame of the CT perfusion sequence.

**Data normalization**    We normalize time values by standardizing to zero mean and unit variance. For attenuation values, we subtract the baseline (first frame), add a +5 HU offset to handle negative values, and divide by the dataset maximum value (170 HU) to scale to [0,1]. During reconstruction, we reverse this process.

## Appendix B. Baseline: Gaussian variate curve fit

Following principles from the indicator dilution theory, we implemented a four-parameter gamma variate function to model the contrast agent dynamics:

$$f(t) = a \cdot (t - b)^c \cdot e^{-(t-b)/d} \cdot H(t - b) \tag{3}$$

where $H(t - b)$ is the Heaviside step function. The gamma variate's parameters capture the key physiological aspects of contrast dynamics: amplitude $a$ represents the peak enhancement reflecting maximum contrast concentration, time offset $b$ indicates contrast arrival time in the tissue, shape parameter $c$ characterizes the rise time corresponding to tissue perfusion rate, and scale parameter $d$ describes the washout rate. Parameter bounds and initialization were defined based on the measured intensity values $C(t)$ at the expected domain $t \in [t_0, \ldots, t_T]$:

$$
\begin{aligned}
&0 \leq a \leq 5 \max(C(t)) \\
&t_0 \leq b \leq t_{\max(C(t))} \\
&0.1 \leq c \leq 5.0 \\
&t_T/10 \leq d \leq t_T
\end{aligned}
\tag{4}
$$

Initial parameter estimates were set as:

$$
\begin{aligned}
a_0 &= \max(C(t)) \\
b_0 &= t_0 \\
c_0 &= 1.0 \\
d_0 &= t_T/3
\end{aligned}
\tag{5}
$$

These constraints ensure physiologically plausible fits while providing sufficient flexibility to capture varying perfusion patterns. The initialization strategy proved robust across our dataset. We enforce that the attenuation returns to baseline levels after the contrast passes through.

## Appendix C. Additional phantom data results

Table 3 lists the mean error and standard deviation on the phantom data. We observe a larger variation for the curve-fit baseline than our proposed method.

For the qualitative comparison in Figure 4, we focus on the CBF and TMAX perfusion parameters because these typically assess the acute infarct core and the salvageable tissue. We note that the error tends to increase for the temporal perfusion parameters as we progress to more severe subsampling.

## Appendix D. Inter-vendor results for patient CT perfusion data

Table 4 presents the volumetric differences between STROKEVIEWER and SYNGO.VIA software when measuring infarct core and penumbra tissue volumes with fully sampled CT perfusion data.

Table 3: Mean error (standard deviation) in perfusion parameter estimation from phantom data.

| Scenario | Time pts. | CBF [ml/100g/s] | CBV [ml/100g] | MTT [s] | Tmax [s] | DELAY [s] |
|---|---|---|---|---|---|---|
| $t \in [t_0, t_2, ...]$ | 15/30 | -0.1 (2.4) | 0.0 (0.2) | 0.1 (5.6) | 0.0 (4.8) | -0.2 (4.6) |
| $t \in [t_0, t_4, ...]$ | 7/30 | 5.4 (6.5) | 0.1 (0.3) | -2.8 (6.2) | -1.3 (4.6) | -0.1 (4.2) |
| $t \in [t_0, t_{pa}, t_{pv}, t_{lv}]$ | | | | | | |
| – proposed | 4/30 | 3.6 (4.7) | -0.2 (0.4) | -3.4 (5.7) | -1.3 (4.4) | 0.1 (4.0) |
| – curve fit | 4/30 | 7.9 (18.8) | 0.1 (0.5) | -3.0 (18.4) | -3.8 (17.0) | -3.5 (16.3) |
| $t \in [t_0, t_{pa}]$ | 2/30 | 11.2 (8.9) | 2.6 (1.3) | 0.5 (4.5) | 1.0 (3.0) | 0.4 (2.4) |

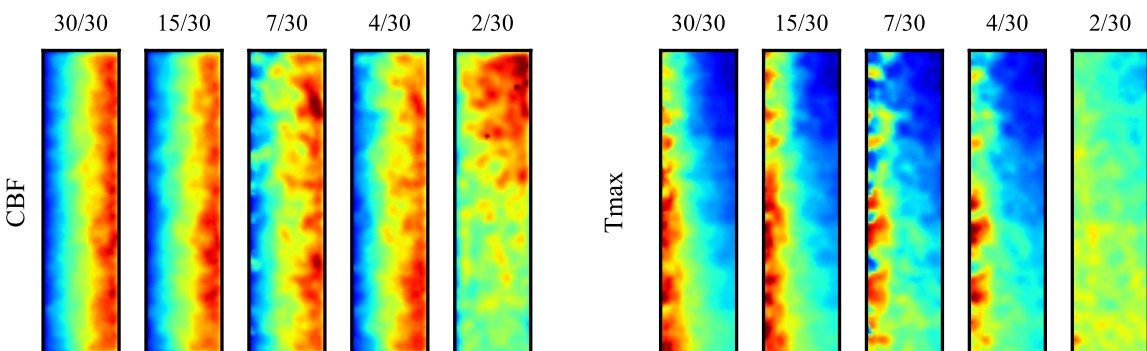

Figure 4: Comparison of commercially available CT perfusion analysis software Syngo.via perfusion estimates (CBF and Tmax) derived from complete CT perfusion data and four subsampling protocols. Maps show phantom data with CBV= 5 ml/100g arranged in a grid where DELAY increases top-to-bottom $(0.0 - 3.0\,\text{s})$ and MTT increases right-to-left $(3.4 - 24.0\,\text{s})$, resulting in CBF increasing left-to-right.

## Appendix E. Perfusion maps for multiphase CT angiography-derived tissue attenuation data

Figure 5 present more qualitative results for perfusion maps derived from multiphase CT angiography data. The perfusion maps from STAR reconstructed curves appear smoother. While we can observe the lower CBF and elevated Tmax regions, the quality of the estimated perfusion maps is not the same as the reference perfusion maps. The elevated Tmax in the same region as the reference maps suggests the quality may be sufficient for detection. However, the accuracy is inadequate for core and penumbra estimates.

Table 4: Infarct core and penumbra volume (ml) estimates based on fully sampled data, analyzed with Syngo.via and StrokeViewer. The table lists median (IQR) (absolute) volumetric difference (VD, AVD) over the test patients between the two softwares. Symbols indicate if smaller (↓) or closer to zero (0) values denote better performance.

| | | StrokeViewer – Syngo.via | |
|---|---|---|---|
| Data | Volume | VD (0) | AVD (↓) |
| CTP | Core | –10.5 (–10.2–(–6.5)) | 10.5 (6.5–20.2) |
| | Penumbra | –88.5 (–107.9–(–62.0)) | 88.5 (62.0–107.9) |

## Appendix F. Latent visualization

Figure 3 visualizes the inferred latent space with each combination of training loss functions, for attenuation curves simulated with MTT = 4 seconds and DELAY = 2 seconds. With the contrastive loss enabled, we see that the latents corresponding to attenuation curves simulated with identical perfusion values are much closer in the latent space. The more compact latent space reduces the likelihood of converging to undesirable local optima.

## Appendix G. Ablation studies

Table 5 lists the results for the ablation studies. We observe that the contrastive loss reduces the error on the perfusion parameters. Moreover, we see that a sampling interval of 8 seconds is a good middle ground in terms of performance, balancing the error in CBF and DELAY.

We conducted empirical tests of inference stability using different hyperparameter settings (learning rates ranging from $10^{-3} - 10^{-1}$, iteration counts from $200 - 2000$, and various loss weightings) on our validation set. The reconstruction quality remained fairly consistent across these settings, likely due to the regularizing effect of the contrastive loss on the latent space.

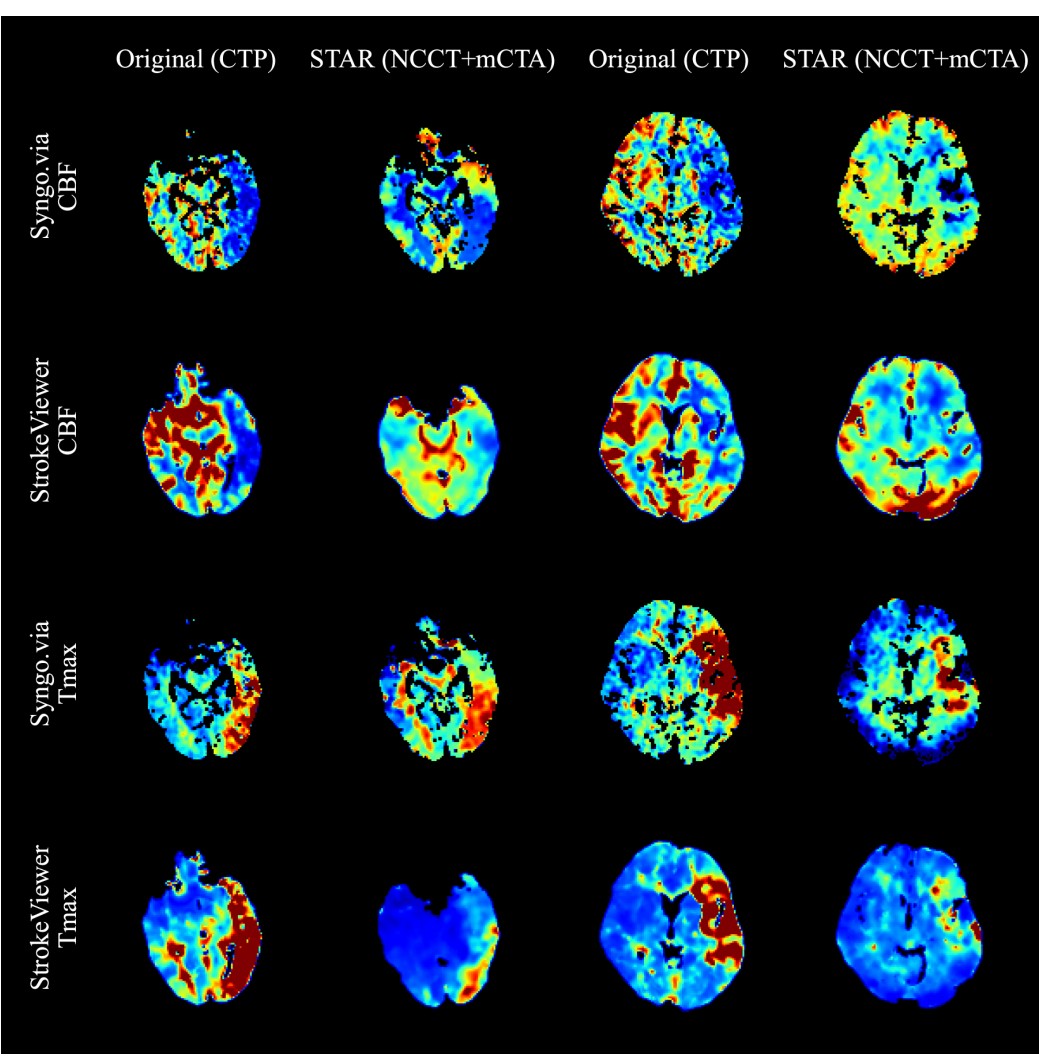

Figure 5: Qualitative results for perfusion maps derived from multiphase CT angiography data.

Table 5: Ablation studies results for different combinations of loss functions and various sampling intervals. Values closer to zero represent better performance.

| $\mathcal{L}$'s used | CBF [ml/100g/s] | CBV [ml/100g] | MTT [s] | TMAX [s] | DELAY [s] |
|---|---|---|---|---|---|
| $\mathcal{L}_{\mathrm{data}}$ | 7.4 | 1.4 | 5.4 | 4.2 | 3.5 |
| $\mathcal{L}_{\mathrm{data}} + \mathcal{L}_{\mathrm{reg}}$ | 5.9 | 0.6 | 4.8 | 2.8 | 2.2 |
| $\mathcal{L}_{\mathrm{data}} + \mathcal{L}_{\mathrm{contrastive}}$ | 3.3 | 0.9 | 4.0 | 1.3 | 0.7 |
| $\mathcal{L}_{\mathrm{data}} + \mathcal{L}_{\mathrm{reg}} + \mathcal{L}_{\mathrm{contrastive}}$ | 4.5 | 0.3 | 3.8 | 1.6 | 0.8 |
| $t_{\mathrm{pa}} \leftrightarrow t_{\mathrm{pv}} \leftrightarrow t_{\mathrm{lv}}$ interval | CBF [ml/100g/s] | CBV [ml/100g] | MTT [s] | TMAX [s] | DELAY [s] |
| 6 sec. | 2.9 | 0.3 | 3.7 | 1.7 | 1.2 |
| 8 sec. | 4.5 | 0.3 | 3.8 | 1.6 | 0.8 |
| 10 sec. | 6.0 | 0.4 | 4.2 | 1.7 | 0.7 |

