# OpenReview forum: "Neural fields for tissue attenuation curve reconstruction in sparsely sampled time-resolved CT"
_MIDL.io/2025/Conference — MIDL 2025 Oral_

### Official Review · Reviewer_1sGY · 2025-02-16

**Confidence:** 4
**Preliminary Rating:** 5
**Recommendation:** Oral
**Final Rating:** 5

**Summary:**

To reduce the radiation dose involved in CT perfusion imaging, the authors of this paper present a novel method based on conditional neural fields, which reconstructs full temporal attenuation curves from a reduced set of temporal acquisitions. Even though the network is trained on synthetic data generated from phantoms, the quantiative and qualitative results on both phantom and patient data are convincing and highlight once more the strength of shallow implicit neural networks.

**Strengths:**

- The paper is very well written, interesting to read, and easy to follow
- The proposed method is novel and it is an elegant solution to reconstruct/interpolate the complete attenuation curves
- The method is thoroughly evaluated and the quantitative and qualitative results are very convincing, especially considering the size of the neural network and complexity of the problem
- The appendix nicely complements the main manuscript

**Weaknesses:**

- Even though the thorough evaluation is one of the paper's strengths, a negative result is the brevity of the results section, which too often only refers to the appendix or figures/tables without providing sufficient interpretation. Consequently, the main manuscript would not be self-contained without reading the appendix.

- The evaluation seems to mainly consider setups where equally-temporally-spaced data is available

**Detailed Comments:**

- Network architecture: The authors note that the network learns the neural field, mapping time to Hounsfield Units. Since the final activation function is a sigmoid, the output seems to be normalized. Could the authors please confirm whether this is the case? If the output is in normalized units, what normalization was applied? Similarly, how is the input represented (is the input the frame ID or in seconds, or something else)?

- While it is great to see the visual impact of the different loss combinations onto the latent space, I believe that the space could be used to deepen the results section and hence I suggest to move the figure into the appendix.

- Page 4, section "training": A) Please clarify in the text that the vector p are the perfusion parameters. B) How were the thresholds selected and how would different values impact the results? C) Were always all temporal measurements of the phantom data used for training?

- Experiments section: To my understanding, the experiments mainly show results for when sparse but equally-spaced observations of the full attenuation curve are available. The authors mention that the proposed method could help with data showing severe motion corruption, possibly by discarding bad frames (?). Could the authors please comment on what they believe happens if one does not take equally spaced measurements? What would be the optimal sampling protocol to boost the performance and limit the radiation exposure?

**Justification Of The Final Rating:**

I would like to thank the authors for the detailed answers to all of my questions and for adding all suggested changes. The revised manuscript is of even higher quality and I am confident that this submission will be of great interest to the community.

**Justification Of The Preliminary Rating:**

The proposed method is novel and elegantly tackles a clinically relevant problem. The experiments are well justified and the quantitative and qualitative results are convincing. Overall I believe that this is a very good paper that will be of great interest to the MIDL community.

**Questions To Address In The Rebuttal:**

In addition to the detailed comments, I would greatly appreciate if the authors could share their thoughts on the following questions:

- During inference the input encoding vector is tuned based on sparse temporal data and then is able to reconstruct a full attenuation curve. Since it is not guaranteed that the network is producing the most accurate solution inbetween the given measurement points, would such a method be of concern when used in the clinical context? Do the authors see any way to increase the trust into this specific method or check whether the predicted attenuation curve is plausible?

- Figure 2, left: While it is great to see that STAR is producing qualitatively similar perfusion maps when giving only 4 frames instead of 30, it would be also very interesting to see the STAR output for 30 frames (and maybe also some intermediate configurations). In particular, I would be interested to understand how well STAR can reconstruct the full attenuation curve. If large differences are still visible at 30 frames, it would be very interesting to understand whether this is a shortcoming of latent encoding vector (or its optimization during inference) or of the network capacity.

**Special Issue:**

Yes

---

> ### Author Response · Authors · 2025-03-06
>
> **Response review 3:**
> We appreciate the time and effort spent reviewing the manuscript and thank the reviewer for their kind words and valuable suggestions for improving the paper.
>
> **Questions to be addressed:**
>
> **DC1:** Network architecture: The authors note that the network learns the neural field, mapping time to Hounsfield Units. Since the final activation function is a sigmoid, the output seems to be normalized. Could the authors please confirm whether this is the case? If the output is in normalized units, what normalization was applied? Similarly, how is the input represented (is the input the frame ID or in seconds, or something else)?
> **ANSWER:**
> For normalization, the input time values (in seconds from 0-58s with 2s intervals) are standardized to zero mean and unit variance. The output attenuation values are indeed normalized 1\) by first removing the baseline of the CTP sequence (i.e. the first frame) 2\) adding a \+5 HU offset to all values (to handle potential negative values) and 3\) dividing by the maximum value in the dataset, scaling everything to [0,1] range. The maximum HU value is 170\. During reconstruction, this process is reversed by multiplying by the maximum value and subtracting the offset and adding the baseline to recover the original scaling.
>
> **CHANGES IN MANUSCRIPT:**
> **In the Appendix Datasets, we add a paragraph “Data normalization”:**
> “We normalize time values by standardizing to zero mean and unit variance. For attenuation values, we subtract the baseline (first frame), add a \+5 HU offset to handle negative values, and divide by the dataset maximum value (170 HU) to scale to [0,1]. During reconstruction, we reverse this process.”
>
> **DC2:** While it is great to see the visual impact of the different loss combinations onto the latent space, I believe that the space could be used to deepen the results section and hence I suggest to move the figure into the appendix.
> **ANSWER:**
> Thank you for the suggestion. With the additional page allowed in the camera-ready version, we can maintain this figure while also including more results from Table 2 in the main manuscript, enriching the results section.
>
> **CHANGES IN MANUSCRIPT:**
> Moved the top part of Table 2 to the main manuscript accompanied with a description in the text.
> In the Results, we add:
> “For reference, Appendix D lists the inter-software differences (StrokeViewer--Syngo.via) for infarct core and penumbra volumes. In the case of fully sampled CT perfusion data, the difference is \--10.5\\,(--10.2--(--6.5)) ml for the infarct core.”
>
> In Appendix D, now called: “Inter-vendor results for patient CT perfusion data” we leave the inter-vendor comparison that originally was in the same table. Accompanied with:
> “Table 4 presents the volumetric differences between StrokeViewer and Syngo.via software when measuring infarct core and penumbra tissue volumes with fully sampled CT perfusion data.”

---

> ### Author Response · Authors · 2025-03-06
>
> **DC3:** Page 4, section "training": A) Please clarify in the text that the vector p are the perfusion parameters. B) How were the thresholds selected and how would different values impact the results? C) Were always all temporal measurements of the phantom data used for training?
>
> **ANSWER A:**
> Thank you for pointing this out. You're correct that p refers to the perfusion parameters (CBF, CBV, MTT, Tmax, delay). We have clarified this in the figure caption and text.
> **CHANGES IN MANUSCRIPT:**
> In the Methods, we revised:
> “For any two curve samples i and j, we measure the similarity of the perfusion parameters p∈ℝ⁵, corresponding to the cerebral blood flow (CBF), cerebral blood volume (CBV), mean transit time (MTT), time-to-maximum (Tmax), and the Delay, that were used to simulate these curves…”
> We also changed the caption of Figure 1 to:
> "Figure 1: Overview of STAR. During training, a conditional neural field is trained on fully sampled simulated CT perfusion data with perfusion parameters p∈ℝ⁵ (CBF, CBV, MTT, Tmax, delay). At inference, only the latent variable is updated based on the observed data on the subsampled domain."
>
> **ANSWER B:**
> The threshold τ=0.1 was an intuitive choice representing approximately 10% difference in normalized parameter space. This value groups physiologically similar patterns together while separating distinct states. The margin m=1.0 ensures adequate separation between dissimilar patterns, but was not motivated on a physiological basis. Our experiments confirmed this approach works well, with the inclusion of contrastive loss itself providing more benefit than specific threshold adjustments. This intuitive threshold helps create a structured latent space that improves reconstruction from sparse measurements. In future work, these parameters could be systematically tuned for potentially even better performance.
> **CHANGES IN MANUSCRIPT:**
> **Added in the discussion:**
> “The threshold τ=0.1 was chosen to represent approximately 10% difference in normalized parameters, which proved effective in practice, while m=1.0 was an empirical choice without specific physiological basis. In future work, these parameters could be systematically tuned for potentially even better performance.”
>
> **ANSWER C:**
> Indeed all temporal measurements of the phantom data were used for training.
> **CHANGES IN MANUSCRIPT:**
> In the Methods, training. we change:
> “During training, we optimize both the neural field parameters θ and the latent variables z⃗ to minimize the difference between our predictions and the training data.”
> to:
> “During training, we optimize both the neural field parameters θ and the latent variables z⃗ to minimize the difference between our predictions and the training data with all 30 time points.”
>
> **DC4:** Experiments section: To my understanding, the experiments mainly show results for when sparse but equally-spaced observations of the full attenuation curve are available. The authors mention that the proposed method could help with data showing severe motion corruption, possibly by discarding bad frames (?). Could the authors please comment on what they believe happens if one does not take equally spaced measurements? What would be the optimal sampling protocol to boost the performance and limit the radiation exposure?
> **ANSWER:**
> We apologize for any confusion. Our experiments don't use equally-spaced measurements \- our protocol uses timepoints at 0s, 16s, 24s, and 32s (uneven intervals of 16s, 8s, 8s). The algorithm is flexible regarding spacing; it can handle any temporal distribution of measurements and therefore can accept data from any CT scanner with any acquisition protocol. Optimal sampling depends on the region of interest: arterial regions benefit from earlier, more frequent sampling, while delayed tissue perfusion requires later time points with wider spacing. We selected our specific protocol because it aligns with multiphase CT angiography, allowing the same model to work for both CTP and mCTA. While our current approach works well, further optimization of sampling patterns is certainly possible in future work.
> **CHANGES IN MANUSCRIPT:**
> In the Experiments, Patient study, we modify:
> “We extract sparse measurements t∈\[t\_0, t\_pa, t\_pv, t\_lv\] from fully sampled CT perfusion data. Since, at inference time, the arterial peak is not available, we use the same time points that we extracted from the phantom data.”
> to:
> “We extract sparse measurements t∈\[t\_0, t\_pa, t\_pv, t\_lv\] from fully sampled CT perfusion data. Our method is flexible regarding temporal spacing and can handle any distribution of measurements, making it compatible with various CT scanner acquisition protocols. Since, at inference time, the arterial peak is not available, we use the same time points that we extracted from the phantom data.”

---

> ### Author Response · Authors · 2025-03-06
>
> **Q1:** During inference the input encoding vector is tuned based on sparse temporal data and then is able to reconstruct a full attenuation curve. Since it is not guaranteed that the network is producing the most accurate solution in between the given measurement points, would such a method be of concern when used in the clinical context? Do the authors see any way to increase the trust into this specific method or check whether the predicted attenuation curve is plausible?
> **ANSWER:**
> Our method can't guarantee 100% accuracy, but the results show that core and penumbra estimates from reconstructed data are comparable to those from fully sampled data, with differences within the range of typical inter-vendor variation.
> The small network with tanh activation functions naturally produces smooth transitions between points, and our contrastive loss ensures physiologically similar curves for similar perfusion parameters. These design choices strongly regularize curve shapes, preventing unrealistic Hounsfield Unit values.
> In the future, spatiotemporal neural fields could further improve trustworthiness by incorporating information from neighboring voxels when predicting tissue attenuation curves.
>
> **CHANGES IN MANUSCRIPT:**
> **Added in Discussion:**
> Future work could incorporate spatial context through neural fields that operate in both spatial and temporal domains (Dupont et al., 2022; Bauer et al., 2023), leveraging brain tissue's spatial coherence where neighboring voxels share similar attenuation patterns.
>
> **Q2:** Figure 2, left: While it is great to see that STAR is producing qualitatively similar perfusion maps when giving only 4 frames instead of 30, it would be also very interesting to see the STAR output for 30 frames (and maybe also some intermediate configurations). In particular, I would be interested to understand how well STAR can reconstruct the full attenuation curve. If large differences are still visible at 30 frames, it would be very interesting to understand whether this is a shortcoming of latent encoding vector (or its optimization during inference) or of the network capacity.
> **ANSWER:**
> When STAR is provided with all 30 frames during inference, it essentially creates an exact copy of the input data. This occurs because our network has sufficient capacity to fit the full temporal sequence perfectly. In this scenario, the perfusion maps generated from STAR's output would be identical to those from the original CT perfusion data, as there is no information loss or approximation happening when we sample the learned network at the exact time points used for training. The latent optimization process simply converges to represent the exact curve being observed. The differences we see with 4 frames arise from the challenge of reconstructing the complete curve from sparse measurements. With all 30 frames, this reconstruction challenge disappears.

---

### Official Review · Reviewer_zUAE · 2025-02-17

**Confidence:** 4
**Preliminary Rating:** 4
**Recommendation:** Poster
**Final Rating:** 4

**Summary:**

CT-Perfusion imaging is used for diagnosing ischemic strokes. A series of around 30 scans are made as a contrast agent passes through the blood vessels of the brain. A point in a blood vessel exhibits a rise and fall of intensity as the contrast agent passes through it. This work models this attenuation curve using an implicit neural field as an autodecoder, conditioned on the patient . Using this approach, this curve can be recovered using fewer scans than classically needed, as the neural field acts as a prior on these curves. This has the potential to massively reduce the dose exposure to patients.

**Strengths:**

This work proposes a novel approach of modeling the brain perfusion using shallow implicit neural fields. Implicit neural fields as autodecoders have been applied before for similar tasks as priors that allow for a good quality recovery with sparse measurements and are in my opinion an solid choice in this application. In addition, a contrastive loss has been applied in the latent space, which has been shown to contribute to the performance in an ablation. The method can be used directly with existing perfusion analysis software which is advantageous for making comparisons, especially as the details of implementation of those is not necessarily known.

**Weaknesses:**

The method was evaluated on phantom- and patient data, but only on a very limited amount. I understand the difficulty of accessing patient data, however I think the results might benefit from including more data. The ISLES 24 dataset (of which data was used in this work) contains to my knowledge a lot more data. Furthermore, the patient data was filtered and resampled to match the phantom data, which could limit the applicability of the method.

More differentiated results would have been appreciated, e.g. with the performance broken down on different types of strokes.

The architectural details of the network used are not motivated and it would be interesting to include some discussion about it, even though it is not the main focus of the work presented.

**Detailed Comments:**

I’d like to elaborate some of the issues I found, and ask the authors to take them into consideration for a final version:

- Table 1 reports MAEs, it would be interesting to also have some measure of spread (e.g. standard deviation) in addition to that.


- Figure 4 in the Appendix shows some 2d plots from a software (“syngo.via”) that I’m not familiar with. Could you please elaborate what is displayed and what the two axes are referring to?

- The overview figure 1 shows that $\vec p \in \mathbb R^5$, as far as I understand this refers to the 5 parameters CBF, CBV, MTT, TMAX, DELAY. It would be nice to mention $\vec p$ and clarify this in the caption or in the text.


Some details on the notation. There are a few things that I would notate differently using - as far as I am aware - more common conventions. I am aware that multiple conventions exist and I want to stress that these are not critical to the content of the paper, but might help the understanding. I would just like you to consider the suggestions and decide whether they would make sense in your perspective too:

- In Figure 1 a calligraphic “R” is used, multiple times to indicate real numbers, I would propose using the “blackboard bold/double struck” symbols instead, as they are used in the text.

- In section 2.1. the function signature is often given along with the argument, like $f_\theta(t) : \mathbb R \to \mathbb R$, I’d write this as $f_\theta : \mathbb R \to \mathbb R$ or $f_\theta: T \to \mathbb R$  and define $T$ separately (otherwise it could  be understood that $f$ returns a *curve* $f(t)$ for each time point $t$). Similarly I’d write $f_\theta: T \times \mathbb R^d \to \mathbb R$ instead of $f_\theta (t, z): \mathbb R^{1+d} \to \mathbb R$.

**Justification Of The Final Rating:**

I would like to thank the authors for their comments and addressing some of the raised issues in the rebuttal. The authors clarified some of the concers and shared their insights into some of the choices they made.

**Justification Of The Preliminary Rating:**

This paper presents a novel application of neural fields for representing and interpolating the attenuation curve of perfusion CTs. It is well motivated, clearly written and shows very promising results. There are a few technical details as well as the small number of patients that limit in my opinion the strength of the current results, however, I am confident that the proposed method is sound and a valuable contribution to the field.

**Questions To Address In The Rebuttal:**

1. You mentioned the inference time, but I’d be curious to know also: How resource intensive is the training as well as the inference in terms of both time and GPU memory?

2. The chosen network is quite small, as of course dictated by the low complexity of this 1D signal. Have you considered other architectures, especially maybe deeper ones? Can you elaborate the decisions that went into the choices for this particular architecture?

3. The curves are mainly characterized using five parameters. Wouldn’t that imply that the latent space could also be reduced to 5 dimensions, given an appropriate network?

4. I criticized the small number of patients data included in the evaluation. Can you comment on this especially in the context of the ISLES24 dataset?

5. The closer two points (in terms of physical distance) are in the brain the more likely they have a curve (unless e.g. one point is in the center of an artery and the other outside of this artery in the parenchyma). Did you consider using this spatial relationship in your model?

6. You report that the latent vector $z$ for the inference is initialized with a normal distribution. Intuitively I would think that it makes more sense to initialize them all exactly the same (e.g. with zeros). Can you comment on this?

**Special Issue:**

No

---

> ### Author Response · Authors · 2025-03-06
>
> **Response review 2:**
> We kindly thank the reviewer for their valuable feedback on our paper. We addressed the specific comments below and adjusted the manuscript where necessary.
>
> **Questions to be addressed:**
> **DC1:** Table 1 reports MAEs, it would be interesting to also have some measure of spread (e.g. standard deviation) in addition to that.
> **ANSWER:**
> We agree with the reviewer. Upon further consideration, we determined that reporting the standard deviation over absolute values is not the most effective way to show spread. Therefore, we added a table with the mean error and standard deviations over the differences to provide more clinically relevant information. We have added this table in the appendix and refer to the table in the main manuscript.
> **CHANGES IN MANUSCRIPT:**
> In the Results, we add:
> “Table 1 lists the mean absolute error. We refer to Table 3 in Appendix C for the mean error en standard deviation. As we subsample more extensively, we see that the errors in the temporal perfusion …”
> In the Appendix. alongside the new table, we add:
> “Table 3 lists the mean error and standard deviation on the phantom data. We observe a larger variation for the curve-fit baseline than our proposed method. For the qualitative comparison in Figure 4, we focus on the CBF…”
>
> **DC2:** Figure 4 in the Appendix shows some 2d plots from a software (“syngo.via”) that I’m not familiar with. Could you please elaborate what is displayed and what the two axes are referring to?
> **ANSWER:**
> Figure 4 displays perfusion maps from our phantom data test set generated using Syngo.via, which is commercial Siemens CT perfusion analysis software commonly used in clinical settings. The figure shows CBF and Tmax parameter maps with constant CBV \= 5 ml/100g, arranged in a grid format where:
>
> - Each row corresponds to different delay values (increasing from top to bottom: 0.0s to 3.0s)
> - Each column corresponds to different MTT values (increasing from right to left: 3.4s to 24.0s), which means CBF (=CBV/MTT) increases from left to right
>
> This grid layout allows quick visual comparison between the reference maps (fully sampled CT perfusion) and those generated from our various subsampling scenarios.
>
> **CHANGES IN MANUSCRIPT:**
> We updated the Caption for Figure 4 with:
> “Figure 4: Comparison of commercially available CT perfusion analysis software Syngo.via perfusion estimates (CBF and Tmax) derived from complete CT perfusion data and four subsampling protocols. Maps show phantom data with CBV=5 ml/100g arranged in a grid where delay increases top-to-bottom (0.0-3.0s) and MTT increases right-to-left (3.4-24.0s), resulting in CBF increasing left-to-right.”
>
> **DC3:** The overview figure 1 shows that p∈ℝ⁵, as far as I understand this refers to the 5 parameters CBF, CBV, MTT, TMAX, DELAY. It would be nice to mention and clarify this in the caption or in the text.
> **ANSWER:**
> Thank you for pointing this out. You're correct that p refers to the perfusion parameters (CBF, CBV, MTT, Tmax, delay). We have clarified this in the figure caption and text.
> **CHANGES IN MANUSCRIPT:**
> In the Methods, we revised:
> “For any two curve samples i and j, we measure the similarity of the perfusion parameters p∈ℝ⁵, corresponding to the cerebral blood flow (CBF), cerebral blood volume (CBV), mean transit time (MTT), time-to-maximum (Tmax), and the Delay, that were used to simulate these curves…”
> We also changed the caption of Figure 1 to:
> "Figure 1: Overview of STAR. During training, a conditional neural field is trained on fully sampled simulated CT perfusion data with simulation parameters p∈ℝ⁵ (CBF, CBV, MTT, Tmax, delay). At inference, only the latent variable is updated based on the observed data on the subsampled domain."
>
> **DC4-5:** In Figure 1 a calligraphic “R” is used, multiple times to indicate real numbers, I would propose using the “blackboard bold/double struck” symbols instead, as they are used in the text. In section 2.1, the function signature…
>
> **ANSWER:**
> Thank you for your careful review and we apologize for the inconsistency. We replaced the calligraphic "R" with the standard blackboard bold/double struck symbol ℝ throughout Figure 1 to match the notation used in the main text. Also, we changed the notation regarding the function signature.
> **CHANGES IN MANUSCRIPT:**
> Updated Figure 1, please see the manuscript with tracked changes.
> Updated function signatures, please see the manuscript with tracked changes.

---

> ### Author Response · Authors · 2025-03-06
>
> **Q1:** You mentioned the inference time, but I’d be curious to know also: How resource intensive is the training as well as the inference in terms of both time and GPU memory?
> **ANSWER:**
> Training takes approximately 22 minutes on a single NVIDIA RTX 2080S GPU with 8GB memory. The model requires only \~1.6GB GPU memory during training with batch size 4000, and \~0.7GB during inference.
> When looking up this specific information, we found a small error in our paper. The paper indicates that we used 1,000 density-sampled curves per batch, but it is actually 4,000. We have modified this in the manuscript.
> **CHANGES IN MANUSCRIPT:**
> In Methods, training:
> "Training takes 22 minutes on consumer hardware with an NVIDIA RTX 2080S GPU and requires \\(\\sim 1.6\\)\\,GB."
>
> In Methods. inference:
> We change:
> “We optimize for 1000 iterations using Adam (\<1 minute on consumer hardware with a NVIDIA RTX 2080S), …”
> to:
> “We optimize for 1000 iterations using Adam (\<1 minute and requiring \~0.7GB), …”
>
> **Q2:** The chosen network is quite small, as of course dictated by the low complexity of this 1D signal. Have you considered other architectures, especially maybe deeper ones? Can you elaborate the decisions that went into the choices for this particular architecture?
> **ANSWER:**
> We experimented with deeper architectures in early development but found that too many layers led to the network focusing on high-frequency noise rather than the fundamental shape of attenuation curves. The chosen small network (single hidden layer) actually helps constrain the solution space toward physiologically plausible curves, improving generalization for this relatively simple 1D signal.
> **CHANGES IN MANUSCRIPT:**
> **In the Discussion, we revised the following paragraph:**
> “We deliberately kept the network architecture small, as our experiments with deeper networks showed that they can fit to high-frequency artifacts rather than the fundamental shape of attenuation curves.
> The contrastive loss provides further regularization of the latent space, ensuring reconstructed curves maintain physiologically plausible shapes even from very sparse measurements – a clear advantage over the gamma-variate model…”
>
> **Q3:** The curves are mainly characterized using five parameters. Wouldn’t that imply that the latent space could also be reduced to 5 dimensions, given an appropriate network?
> **ANSWER:**
> Though the five perfusion parameters suggest a 5-dimensional latent space might work (perhaps with a larger network), our 32-dimensional space has more flexibility to capture complex relationships between parameters. While we didn't experiment with smaller latent spaces since it is already relatively small, this would be interesting for future work.
>
> **CHANGES IN MANUSCRIPT:**
> In the Discussion, we add:
> “While the five standard perfusion parameters suggest a minimum dimensionality for the latent space, our larger latent space provides flexibility to capture complex, non-linear relationships between parameters.”
>
> **Q4:** I criticized the small number of patients data included in the evaluation. Can you comment on this especially in the context of the ISLES24 dataset?
> **ANSWER:**
> We faced practical challenges with the ISLES24 dataset. To use commercial perfusion software, we had to convert the NIfTI files to DICOM format because Strokeviewer and Syngo.via do not allow for using Nifti images as input. This only worked for cases with exactly 16 slices that matched our phantom structure of 16 slices. Plus, processing each case in StrokeViewer and Syngo.via was manual and time-consuming, which is why we did thorough quantitative testing with the phantom (where we have ground truth values) and focused qualitative validation with patient data. The results were consistent across both, which gives us confidence in our method. However, we have acknowledged this limitation in the manuscript.
> **CHANGES IN MANUSCRIPT:**
> **In the Discussion, we add:**
> “Our validation cohort is limited in size, which represents a current limitation to be addressed in future work with larger clinical datasets.”

---

> ### Author Response · Authors · 2025-03-06
>
> **Q5:** The closer two points (in terms of physical distance) are in the brain the more likely they have a curve (unless e.g. one point is in the center of an artery and the other outside of this artery in the parenchyma). Did you consider using this spatial relationship in your model?
> **ANSWER:**
> You raise an excellent point about spatial relationships in brain tissue. We did consider incorporating spatial context, but decided to focus first on proving the temporal reconstruction concept. As noted in our discussion, extending to spatial domains would require more complex neural field architectures. Unfortunately, conditional neural fields with global conditioning scale poorly with increased dimensionality. Moreover, our phantom's artificial arrangement (curves in a block-structure) doesn't reflect actual brain anatomy, limiting the benefits of spatial modeling with our current phantom. Incorporating spatial relationships would benefit most from training on real patient data, with accurate spatial relationships between the curves, or a phantom which reflects such spatial relationships.
> For future work, we aim to explore spatial-aware models trained on patient data, potentially using equivariant neural fields that can better handle the local spatial coherence you describe, while maintaining computational efficiency.
>
> **CHANGES IN MANUSCRIPT:**
> We modified the following paragraph in the Discussion:
> “STAR only considers the temporal domain for reconstruction. Future work could incorporate spatial context through neural fields that operate in both spatial and temporal domains (Dupont et al., 2022; Bauer et al., 2023). However, for more complex spatio-temporal signals, neural fields with global conditioning scale poorly (Dupont et al., 2022; Bauer et al., 2023; Papa et al., 2024; Xie et al., 2022). Recent work on equivariant neural fields could provide a solution with a geometry-informed latent space (Wessels et al., 2024).”
> to:
> “STAR only considers the temporal domain for reconstruction. Future work could incorporate spatial context through neural fields that operate in both spatial and temporal domains (Dupont et al., 2022; Bauer et al., 2023), leveraging brain tissue's spatial coherence where neighboring voxels share similar attenuation patterns. However, for more complex spatio-temporal signals, neural fields with global conditioning scale poorly (Dupont et al., 2022; Bauer et al., 2023; Papa et al., 2024; Xie et al., 2022). Recent work on equivariant neural fields could provide a solution with a geometry-informed latent space (Wessels et al., 2024).”
>
> **Q6:** You report that the latent vector  for the inference is initialized with a normal distribution. Intuitively I would think that it makes more sense to initialize them all exactly the same (e.g. with zeros). Can you comment on this?
> **ANSWER:**
> Random Gaussian initialization of latent vectors offers several advantages over zero initialization:
> 1\. It enables better exploration of the latent space during optimization
> 2\. It introduces asymmetry at initialization, preventing all neurons from updating identically and allowing diverse features to emerge more quickly
> 3\. It helps avoid saddle points, where gradients are near-zero in all directions, near the origin of the latent space. High-dimensional spaces (like our latent space) typically have many saddle points, especially near the origin, which can trap optimization algorithms. Starting away from these regions ensures more effective gradient descent.
>
> Our approach follows established practices in neural fields, particularly the DeepSDF work which demonstrated this initialization strategy's effectiveness for representing complex signals with latent-conditioned / autodecoder networks.
>
> **CHANGES IN MANUSCRIPT:**
> **We revised the Methods to include:**
> “All latent variables \\(\\vec{z}\\in\\mathbb{R}^d\\) are randomly initialized from \\(\\mathcal{N}(0, \\frac{1}{\\sqrt{d}})\\), where \\(d=32\\) (Park et al.), which promotes effective optimization and avoids regions with vanishing gradients.”

---

> > ### Comment · Reviewer_zUAE · 2025-03-06
> >
> > I would like to thank the authors for the thorough responses and for addressing the points appropriately, and of course for sharing their insights.
> > I am particularly glad that you decided to add the information about the resource consumption, as this highlights the technical effectiveness of the method. Furthermore also wanted to thank you for updating the caption of Figure 4, with which I know clearly understand the plots, and of course also for the updated tables.
> > Regarding Q6 From my experience with similar problems I must admit that I still not think that a random initialization would be necessary for this particular problem - but then again I also do not expect it to degrade the performance. It was in any case more detail question about the intuition and not crucial. Thank you very much for sharing your insights!

---

### Official Review · Reviewer_LQWv · 2025-02-22

**Confidence:** 5
**Preliminary Rating:** 3
**Final Rating:** 4

**Summary:**

This work presents conditional neural fields to resconstruct tissue attenuation curves from sparse observations for CT perfusion imaging. Different sampling rates were experimented and the results demonstrate that only 4 out of 30 obersavations could generate reasonably good quality CT perfusion images. Further, additional and ablation studies were conducted to demonstrate the performance of the proposed method.

**Strengths:**

1) The problem motivation, research objectives, proposed method, datasets and experiments have been presented clearly.
2) Conditioning the nerual field with latent variable for the task at hand seems interesting to me.
3) Ablation and additional studies have been presented clearly.

**Weaknesses:**

1) The related works do not mention about low-dose CT perfusion imaging.
2) The estimatred Tmax parameter is noisy (Fig. 2) and also CBF & Tmax derived from CT angiography data do not agree with original CTP reconstruction (Fig. 5).
3) Hyperparameters during inference might lead to uncertain estimates.!

**Detailed Comments:**

Please refer to strengths and weaknesses.

**Justification Of The Final Rating:**

The authors have provided a detailed response to the concerns raised. The review comments have been addressed to the satisfaction of the reviewers. The revised manuscript demonstrates improved readability compared to the previous version, and I am pleased to enhance the manuscript's score.

**Justification Of The Preliminary Rating:**

The idea of conditional neural fields for sparse CT perfusion imaging seems interesting. The technical presentaion and the experiments forms the strength of the work. However, small deviation in CT perfusion parameters can significantlty affect the outcomes in clinical scenarios like stroke assessment. I would request authors to re-assess the quality of CBF & Tmax reconstructions and provide a detailed discussion on trade-off between the reconstruction quality and the affect of input radiation dose.

**Questions To Address In The Rebuttal:**

1) A detailed discussion on the points mentioned in weaknesses.
2) During inference does $L_{data}$ and $L_{reg}$ weighted similar to training ?
3) During training $L2$ optimization was used. However during inference $L1$ was used ?
4) Authors are strongly encouraged to add colorbars to all figures with perfusion maps.

---

> ### Author Response · Authors · 2025-03-06
>
> **Response review 1:**
> We would like to thank the reviewer for their kind feedback and suggestions on our manuscript.
>
> **Questions to be addressed:**
>
> **W1:** The related works do not mention about low-dose CT perfusion imaging.
> **ANSWER:**
> Thanks for the suggestion, we added a discussion about low-dose CT perfusion imaging in the discussion.
> **CHANGES IN MANUSCRIPT:**
> In the Discussion, we add:
> “While our paper focuses on temporal undersampling, we acknowledge other dose reduction strategies exist, including hardware-based approaches (Lira et al., 2015), iterative reconstruction algorithms (Rapalino et al., 2012), and denoising techniques (Chen et al., 2017), which could be combined with our method for greater reduction.”
>
> Lira D, Padole A, Kalra MK, Singh S. Tube potential and CT radiation dose optimization. AJR Am J Roentgenol. 2015 Jan;204(1):W4-10. doi: 10.2214/AJR.14.13281. PMID: 25539272\.
>
> Rapalino O, Kamalian S, Kamalian S, Payabvash S, Souza LC, Zhang D, Mukta J, Sahani DV, Lev MH, Pomerantz SR. Cranial CT with adaptive statistical iterative reconstruction: improved image quality with concomitant radiation dose reduction. AJNR Am J Neuroradiol. 2012 Apr;33(4):609-15. doi: 10.3174/ajnr.A2826. Epub 2011 Dec 29\. PMID: 22207302; PMCID: PMC8050448.
>
> Chen H, Zhang Y, Kalra MK, Lin F, Chen Y, Liao P, Zhou J, Wang G. Low-Dose CT With a Residual Encoder-Decoder Convolutional Neural Network. IEEE Trans Med Imaging. 2017 Dec;36(12):2524-2535. doi: 10.1109/TMI.2017.2715284. Epub 2017 Jun 13\. PMID: 28622671; PMCID: PMC5727581.
>
> **W2:** The estimated Tmax parameter is noisy (Fig. 2\) and also CBF & Tmax derived from CT angiography data do not agree with original CTP reconstruction (Fig. 5).
>
> **ANSWER:**
> We agree that the Tmax parameter shows some noise in our reconstructions. This is expected given the aggressive temporal subsampling we are using (only 4 out of 30 timepoints). We acknowledge this limitation in the first point of our Discussion section, noting that sparse sampling naturally misses exact bolus arrival times and wash-out characteristics. For multiphase CT angiography data, we present these limitations \- while we can detect infarcts qualitatively, there are differences in acquisition protocols that affect the precision of measurements. Here we also state that while the presented method can detect infarcts qualitatively, "the perfusion maps lack the precision needed for accurate core-penumbra volume calculations". For CTP, our results show that even with these limitations, core volume estimates remain comparable to the inter-software variability commonly seen in clinical practice.
> **CHANGES IN MANUSCRIPT:**
> In the Discussion, after "The deviations increase with more aggressive sub-sampling protocols." we add:
> "Despite these temporal deviations, our method maintains sufficient clinical value."
>
> **W3:** Hyperparameters during inference might lead to uncertain estimates.
> **ANSWER:**
> We understand  the reviewer's concern about hyperparameter sensitivity during inference. In our experiments, we found that the inference is actually quite robust to hyperparameter settings. The latent space regularization through contrastive loss ensures that optimized latent variables converge to similar regions for similar input data, making the process stable. We've empirically tested different learning rates, iteration counts, and loss weightings and observed consistent reconstruction quality using the 25% of the phantom data (the validation set).
> **CHANGES IN MANUSCRIPT:**
> In appendix G "Ablation studies", we add:
> “We conducted empirical tests of inference stability using different hyperparameter settings (learning rates ranging from 10^-3 \- 10^-1, iteration counts from 200-2000, and various loss weightings) on our validation set. The reconstruction quality remained fairly consistent across these settings, likely due to the regularizing effect of the contrastive loss on the latent space."
>
>
> **Q2:** During inference does L\_data  and L\_reg weighted similar to training ?
> **ANSWER:**
> During inference, we use the same weights as during training (100 and 0.1 respectively), but we set the weight of L\_contrastive to 0 since this loss requires known perfusion parameters which are unavailable at inference time.
> **CHANGES IN MANUSCRIPT:**
> In the methods, we change:
> "We optimize the latents z with L\_data and L\_reg using only the observed data with an ℓ₁-loss, while keeping the neural field parameters θ fixed."
> To:
> "We optimize the latents z with L\_data and L\_reg with the same weights as during training, using only the observed data while keeping the neural field parameters θ fixed. We use L1-loss during inference (rather than the L2-loss used during training) to better preserve the tail characteristics of the attenuation curves."

---

> > ### Comment · Reviewer_LQWv · 2025-03-11
> >
> > I would like to thank authors for the detailed respose to the raised concerns. The revised manuscript demonstrates improved readability compared to the previous version, and I am pleased to enhance the manuscript's score.

---

> ### Author Response · Authors · 2025-03-06
>
> **Q3:** During training L2 optimization was used. However during inference L1 was used ?
> **ANSWER:**
> Yes, we use L2 loss during training but L1 loss during inference. L2 loss during training provides stronger gradients for larger errors and a smoother optimization landscape, making it effective to quickly learn the overall shape patterns across our training set. However, with L2 loss during inference, errors at peak values dominated the optimization, causing reconstructed curves to have insufficient width. By switching to L1 loss for inference, we reduced the disproportionate influence of peak errors, resulting in better preservation of the curve's temporal characteristics, especially the tail portions that are crucial for accurate mean transit time and time-to-maximum measurements.
> **CHANGES IN MANUSCRIPT:**
> In the Methods, we add:
> “We use L1-loss during inference (rather than the L2-loss used during training) to better preserve the tail characteristics of the attenuation curves.”
>
> **Q4:** Authors are strongly encouraged to add colorbars to all figures with perfusion maps.
> **ANSWER:**
> We deliberately omitted colorbars to avoid confusion in the figures, as the two software packages use different units and scales (Syngo.via: physical units with CBF range 0-100 ml/100g/min, Tmax 0-8 seconds; StrokeViewer: relative units with CBF range 0-170%, Tmax 0-8 seconds). We believe that adding colorbars would make the figures more cluttered when the main purpose is qualitative comparison. However, if deemed essential, we'd be happy to add a figure with colorbars in the appendix of the camera-ready version.
>
> **Q5:** I would request authors to re-assess the quality of CBF & Tmax reconstructions and provide a detailed discussion on trade-off between the reconstruction quality and the affect of input radiation dose.
> **ANSWER:**
> We acknowledge there's a trade-off between reconstruction quality and radiation dose reduction. Our approach aligns with the ALARA principle (As Low As Reasonably Achievable), where reducing radiation exposure is beneficial to patients as long as diagnostic quality is maintained. While our reconstructions show some expected deviations in temporal parameters like Tmax, our results demonstrate that the volumetric measurements remain clinically comparable to those derived from fully-sampled data, with differences within the range of inter-software variability seen in clinical practice. For multiphase CT angiography, our method provides a benefit by enabling perfusion analysis where none would otherwise be possible. Many centers choose mCTA over CTP specifically to reduce radiation exposure, accepting the loss of perfusion data. Our approach allows these centers to gain perfusion information without changing their acquisition protocols or increasing radiation dose.
>
> **CHANGES IN MANUSCRIPT:**
> In the Discussion, we add:
> “While Tmax shows expected deviations with 87% fewer acquisitions, the core and penumbra volume measurements remain comparable to fully-sampled data, with differences within inter-software variability ranges. For centers using multiphase CT angiography, our method enables perfusion analysis that would otherwise be unavailable, supporting both radiation safety and diagnostic needs.”

---

### Author Rebuttal · Authors · 2025-03-06

**Rebuttal:**

We have uploaded a new PDF with tracked changes and have answered all the questions of the reviewers. We would like to thank the reviewers and chairs for their time and efforts to evaluate our manuscript.

**Supporting Material:**

/attachment/26688075f3b107093864a026b3f055180f1a7275.pdf

---

### Meta-Review · Area_Chair_zcKB · 2025-03-18

**Recommendation:** Accept (Oral)
**Confidence:** 5

**Metareview:**

All reviewers appreciate the novelty and effectiveness of using implicit neural fields for sparsely sampled CT reconstruction. I recommend acceptance with an oral presentation for better exposure of this novel work to the community.